# Molecular Detection and Phylogeny of *Anaplasma phagocytophilum* and Related Variants in Small Ruminants from Turkey

**DOI:** 10.3390/ani11030814

**Published:** 2021-03-14

**Authors:** Münir Aktaş, Sezayi Özübek, Mehmet Can Uluçeşme

**Affiliations:** Department of Parasitology, Faculty of Veterinary Medicine, University of Firat, 23119 Elazig, Turkey; sozubek@firat.edu.tr (S.Ö.); mculucesme@firat.edu.tr (M.C.U.)

**Keywords:** tick-borne fever, *Anaplasma phagocytophilum*-like 1, PCR-RFLP, small ruminant

## Abstract

**Simple Summary:**

We explored the existence of *Anaplasma phagocytophilum* and related variant in samples of goats and sheep obtained from Antalya and Mersin provinces, representative of Mediterranean region of Turkey. Based on *16S rRNA* and *groEL* genes of *A. phagocytophilum* and related variants, we examined blood samples by polymerase chain reaction (PCR) followed by sequencing. The results showed that the prevalence of *A. phagocytophilum* and *A. phagocytophilum*-like 1 infection was 1.4% and 26.5%, respectively. Sequencing confirmed molecular data and showed the presence of *A. phagocytophilum* and *A. phagocytophilum*-like-1 variant in the sampled animals.

**Abstract:**

*Anaplasma phagocytophilum* causes tick-borne fever in small ruminants. Recently, novel *Anaplasma* variants related to *A. phagocytophilum* have been reported in ruminants from Tunisia, Italy, South Korea, Japan, and China. Based on *16S rRNA* and *groEL* genes and sequencing, we screened the frequency of *A. phagocytophilum* and related variants in 433 apparently healthy small ruminants in Turkey. *Anaplasma* spp. overall infection rates were 27.9% (121/433 analyzed samples). The frequency of *A. phagocytophilum* and *A. phagocytophilum*-like 1 infections was 1.4% and 26.5%, respectively. No *A. phagocytophilum*-like 2 was detected in the tested animals. The prevalence of *Anaplasma* spp. was comparable in species, and no significant difference was detected between sheep and goats, whereas the prevalence significantly increased with tick infestation. Sequencing confirmed PCR-RFLP data and showed the presence of *A. phagocytophilum* and *A. phagocytophilum*-like-1 variant in the sampled animals. Phylogeny-based on *16S rRNA* gene revealed the *A. phagocytophilum*-like 1 in a separate clade together with the previous isolates detected in small ruminants and ticks. In this work, *A. phagocytophilum*-like 1 has been detected for the first time in sheep and goats from Turkey. This finding revealed that the variant should be considered in the diagnosis of caprine and ovine anaplasmosis.

## 1. Introduction

*Anaplasma phagocytophilum* is the agent of tick-borne fever (TBF) or pasture fever, a disease affecting some species of domestic ruminants (cattle, sheep, goats). The bacterium is a pathogenic species for livestock such as ruminants as well as humans in temperate and tropical countries [1,2,3,4]. *Anaplasma phagocytophilum* is transmitted by *Ixodes* spp. and infects host neutrophils and monocytes, where reproduction occurs [1,5]. *Anaplasma phagocytophilum* infection is known as pasture fever and characterized by fever, anorexia, lateral recumbency, dullness, and loss of milk yield in affected hosts [2,4,6].

Increased attention to *A. phagocytophilum* reveals new information about the genetic diversity of the pathogen. Recently two *Anaplasma* variants related to *A. phagocytophilum* have been documented in cattle, sheep, goats, and ticks [7,8,9]. In Japan, *A. phagocytophilum*-like 1 has been detected in deer and *Hemaphysalis longicornis* [10], cattle [11], *Ixodes* spp. [12], and *Haemaphysalis megaspinosa* [13]. *A. phagocytophilum*-like 2 has been identified in *Hyalomma asiaticum* [14], sheep and goats from China [15]. Recently those *Anaplasma* variants have been documented in ruminants from Tunisia [7,8], South Korea [16], and Italy [17].

Various *Anaplasma* species including *A. phagocytophilum* have been documented in ruminants and ticks in Turkey [5,6,18,19,20,21]. However, until now no data on *A. phagocytophilum* variants is available in Turkey. In the current study, 16S rRNA, groEL (heat shock protein) PCR and sequencing were performed to identity *A. phagocytophilum* and *A. phagocytophilum*-like variants in small ruminants from sampling sites in Antalya and Mersin provinces, where the representative Mediterranean area of Turkey.

## 2. Materials and Methods

### 2.1. Study Region and Sample Collection

This survey was conducted in small ruminants farmed in three districts (Alanya, Akseki, Manavgat) from Antalya (latitude 36° 53′ N, longitude 30° 42′ E) and two districts (Anamur, Bozyazı) from Mersin (latitude 36° 47′ N, longitude 34° 37′ E) provinces of Turkey (Figure 1). This area has a Mediterranean climate, with hot humid summers and warm rainy winters. The goats and sheep are kept in closed areas in villages near to the coast during the winter months, and they are taken to the plateaus in the Taurus Mountains in the early spring and grazed in the pastures here until autumn.

The sample size was calculated using the online tool Sample Size Calculator (www.calculator.net/sample-sizecalculator.html, accessed on 1 February 2019), for a confidence level (CL) of 95%, an error margin of 5%. According to this, during April–July 2019, a total of 433 apparently healthy small ruminants (296 goats, 137 sheep) were included in the survey. Blood samples were drawn from the punctured jugular vein into anticoagulated (K3-EDTA) vacutainer tubes and stored at −20 °C freezer until DNA extraction. The goats and sheep were also checked for tick infestations, and a total of 1475 ticks were removed. The collected ticks were preserved in 70% ethanol in Eppendorf tubes. They were identified using taxonomic keys [22]. The animals were grouped into categories according to species (goat and sheep) and the presence of ticks (yes/no). This study secured the approval of the Elazig Veterinary Control Institute (number: 2018/02).

### 2.2. DNA Extraction and Amplification of 16S rRNA Gene

DNA was isolated from 200 µL volumes of whole blood using a DNAeasy Blood Minikit according to the vendor’s recommendations. Genomic DNA from blood of clinically infected cattle with *A. phagocytophilum* [6] was used for positive control. *Anaplasma phagocytophilum*-like variants DNAs, received from Alberto Alberti (University of Sassari, Sassari, Italy) were used as positive controls.

To detect *A. phagocytophilum* and *A. phagocytophilum*-like variants, a nested 16S rRNA PCR was carried out described by Kawahara et al. [10]. The PCR reaction conditions were made according to the previously described studies [10,21]. The nested amplicons were examined by 1.5% agarose gel electrophoresis and visualized using the gel Documentation System (Vilber Lourmat, Marne La Vallee Ceedex, France).

### 2.3. Restriction Fragment Length Polymorphism (RFLP)

*Xcm*I and *Bsa*I restriction enzymes allow the specific discrimination amongst *A. phagocytophilum* and related variants [8,17]. For differentiation of *A. phagocytophilum* and related variants, the nested amplicons obtained in this study were digested with the *Xcm*I and *Bsa*I restriction enzymes as previously described [8,17].

### 2.4. GroEL PCR

To confirm the results of the RFLP assay, the positive samples were screened by a groEL nested PCR for the amplification of *A. phagocytophilum* [23]. The semi-nested PCR reported by Ybañez et al. [24] with the primers EEGro1F/AnaGroe712R and AnaGroe240F was utilized for amplifying of *A. phagocytophilum*-like 1 *groEL* gene. Oligonucleotide primers used in this study were presented in Table 1.

### 2.5. Sequencing and Phylogenetic Analyses

*Anaplasma phagocytophilum* (*n* = 6) and *A. phagocytophilum*-like 1 (*n* = 10) positive PCR amplicons were purified using the QIAquick PCR Purification Kit (Qiagen, Hilden, Germany). The purified amplicons were sent to BM Labosis (Ankara, Turkey) for Sanger sequencing to determine DNA sequences of the *16S rRNA* gene. Multiple alignments were performed with the CLUSTAL Omega ver. 1.2.1 (https://www.ebi.ac.uk, accessed on 1 February 2019). The representative sequences have been submitted to the GenBank (MT881655 and MT881656 for *16S rRNA* gene of *A. phagocytophilum*-like 1 and *A. phagocytophilum*, respectively). The sequence alignment was performed using MUSCLE in Geneious prime [25].

Phylogenetic analyses of the 16S rRNA sequences obtained in this work and the other sequences submitted to GenBank were carried out. The maximum likelihood analysis (ML) carried out in Mega X [26] was utilized to determine the phylogenetic relationship of the *Anaplasma* spp. To sequence evolution, best-fit model was assessed as TN93+G+I by using the jModel test v.0.1.1 [27]. Reliability of internal branches of the tree was evaluated by the bootstrapping method with 1000 iterations [28].

### 2.6. Statistical Analysis

Association of the presence of *Anaplasma* spp. with host species and presence of tick was performed with Epi Info 6.01 (CDC, Atlanta), using the χ2 test and Fisher’s exact test.

## 3. Results

### 3.1. Tick Infestation

Of the 433 small ruminants examined, 190 (43.9%) were infested with at least one tick species. A total of 1475 adult ticks (449 females, 1026 males) were collected from goats (1409/1475, 95.5%) and sheep (66/1475, 4.5%). Six tick species were identified among all collected ticks. *Rhipicephalus bursa* (1269/1475, 86%) was the dominant tick species, followed by *R. turanicus* (98/1475, 6.6%), *Dermacentor marginatus* (94/1475, 6.4%), *Hyalomma marginatum* (8/1475, 0.5%), *R. sanguineus* s.l. (5/1475, 0.3%), and *Ixodes ricinus* (0.06%, only one specimen). The goats were infested with all the identified tick species, whereas sheep were infested with *R. bursa* and *R. turanicus.*

### 3.2. Prevalence and Distribution of Anaplasma spp.

The prevalence of *A. phagocytophilum* and related variants in sampled goats and sheep is presented in Table 2. Overall, 121/433 (27.9%) samples collected in studied regions tested positive for *Anaplasma* spp. by 16S rRNA PCR. The infection rate in goats and sheep was determined as 28% and 27.7%, respectively. RFLP revealed the prevalence of *A. phagocytophilum* and *A. phagocytophilum*-like 1 as 1.4% and 26.5%, respectively. No PCR amplicons derived from goats and sheep were digested by the *Bsa*I enzyme, confirming the absence of Chinese variant (*A. phagocytophilum*-like 2). Of the 121 positive samples with 16S rRNA PCR, 110 (95.6%) were positive with groEL nested PCR. Six of them (6/110, 5.4%) were positive for *A. phagocytophilum* and 104 (94.5%) were positive for *A. phagocytophilum*-like 1 (Table 2).

Association of the frequency of *A. phagocytophilum* and *A. phagocytophilum*-like 1 variant in small ruminants with species and tick infestation is documented in Table 3. The prevalence of *Anaplasma* spp. was comparable in species, and no difference was detected between infection rates in sheep and goats (*p* = 0.9603). However, the prevalence significantly increased with tick infestation in small ruminants (*p* = 0.0003) (Table 3).

### 3.3. Molecular and Phylogenetic Analyses

To validate the RFLP results and identify genetic variants of *A. phagocytophilum*-like 1, randomly selected 10 representative samples were sequenced. The sequences shared 100% identity to each other. Therefore, one representative sequence for *A. phagocytophilum*-like 1 was submitted to the NCBI GenBank database, and deposited with accession number MT881655. This finding indicated that one variant was identified, and named as Aplike1OvineCaprine in this work. BlastN analysis demonstrated that the Aplike1OvineCaprine variant indicated high similarity (99–100%) to those *Anaplasma* isolates deposited in the GenBank as uncultured *Anaplasma* sp. and *A. phagocytophilum*. Moreover, the Aplike1OvineCaprine variant was 100% identical to those of *A. phagocytophilum*-like 1 detected in sheep (Aplike1Ov1, KX702978) and goat (Aplike1GGo2, KM285227) from Tunisia, and cattle from Turkey (Aplike1Bv, MT338494) (Table 4). The *A. phagocytophilum* Akseki11 Sheep Turkey isolate obtained in this study shared 99.3–99.6% identity isolated from *Niviventer confucianus* (*A. phagocytophilum* ZJ-HGA strain, DQ458805) and human (*A. phagocytophilum* HZ strain, NR_074113), respectively.

Phylogenetic analysis using the *16S rRNA* gene showed that our variant (Aplike1OvineCaprine) clustered a distinct group with those of *A. phagocytophilum*-like 1 previously published sequences reported in sheep, goats, cattle, deer, and *Haemaphysalis ginghaiensis* (Figure 2).

## 4. Discussion

*Anaplasma phagocytophilum* causes tick-borne fever in small ruminants and granulocytic anaplasmosis in horses and dogs [1,2]. It is an emerging tick-borne pathogen for humans as well [3]. The genetic diversity of *A. phagocytophilum* is much greater than expected. Indeed, recent studies have revealed the existence of two distinct *Anaplasma* species or variants related to *A. phagocytophilum,* one in Japan and the other in China [11,12,13,14,24]. Then, these pathogens were designated as *A. phagocytophilum*-like 1 and *A. phagocytophilum*-like 2 variants [7,8]. More recently, both genotypes have been documented in ruminants and *R. turanicus* in Tunisia [7,8,17,29], cattle in South Korea [16], and small ruminants in Italy [17]. In the present study, a survey was carried out to detect and identify *A. phagocytophilum* and *A. phagocytophilum*-like variants in small ruminants from the Mediterranean region of Turkey. Our findings provide molecular evidence for the presence of *A. phagocytophilum* and *A. phagocytophilum*-like 1 in sampled sheep and goats. In the previous studies carried out in Turkey, *A. phagocytophilum* has been reported in small ruminants [18,21,30]. However, this is the first time that *A. phagocytophilum*-like 1 variant in sheep and goats have been reported in the country.

Contrary to *A. phagocytophilum*, it has been suggested that both Japanese and Chinese variants do not cause clinical infection in ruminants [8,17]. In this study, a high prevalence for *A. phagocytophilum*-like 1 variant was determined (26.5%), but no clinical infection for tick-borne fever was observed in sheep and goats during sample collection. This result is consistent with the previous suggestions that *A. phagocytophilum*-like variants are considered non-pathogenic for ruminants [8,16,17]. The prevalence of *A. phagocytophilum*-like 1 (26.5%) in small ruminants obtained in this study was higher than that observed in Tunisian sheep (7%) and goats (13.1%) [8], however, it was lower than that observed in other studies conducted in Mediterranean small ruminants (122/203, 60%) from Tunisia and Italy [17].

It has been previously suggested that serological cross-reactions occur between *A. phagocytophilum* and other *Anaplasma* species [31,32]. The same situation may be true in some circumstances for molecular markers, for example a pair of primers (SSAP2f/SSAP2r) based on the *16S rRNA* gene of *A. phagocytophilum* were designed for the specific amplification [10]. However, it has been shown that these primers also detect distinct *Anaplasma* variants related to *A. phagocytophilum* [7,8,24]. In this work, the frequency of pathogenic *A. phagocytophilum* was 1.4%, which is not consistent with the previous studies in Turkey that reported values of 66.7% in Central Anatolia [30] and 19.7% in Eastern Anatolia [21]. The high infection rates obtained in the previous studies may be due to the selected primers for the amplification of *A. phagocytophilum*. EE1/EE2 and SSAP2f/SSAP2r primers have been selected to detect *A. phagocytophilum* in the studies conducted in Central Anatolia [30] and Eastern Anatolia [21], respectively. However, the EE1/EE2 primers are universal for the detection of all *Anaplasma* spp. including *A. phagocytophilum*-like variants [33]. It has been also reported that the SSAP2f/SSAP2r can amplify not only *A. phagocytophilum*, but also *A. phagocytophilum*-like variants [7,8,24]. This study provides molecular data for the circulation of *A. phagocytophilum* and *A. phagocytophilum*-like 1 Turkish small ruminants. Therefore, cross-reactivity between *A. phagocytophilum* and related variant should be considered in interpreting the findings of surveys to be carried out in the area, where *A. phagocytophilum* and *A. phagocytophilum*-like variant co-exist.

As several domestic and wild mammals are hosts or reservoirs for *A. phagocytophilum* [1,2], abundance and intensity of the tick vector, *I. ricinus* in Europe including Turkey are considered a major factor affecting the distribution of the pathogen in a specific area. It is well known that there is no *I. ricinus* in the Eastern and Central Anatolian regions of Turkey [34]. It has been reported that *A. phagocytophilum* is transmitted by *Ixodes* spp. (*I. persulcatus*, *I. scapularis* and *I. ricinus*) in some parts of the world including in Europe [1,35]. In Turkey, *I. ricinus* collected from humans were positive for *A. phagocytophilum* [5]. So far, data on the transmission of *A. phagocytophilum*-like variants by ticks are lacking. A recent study reported that *R. turanicus* was common in sampled sheep and goats in Tunisia, and one *R. turanicus* tick feeding on the goat was found to be infected with *A. phagocytophilum*-like 2 [28]. In the present study, potential vectors of *A. phagocytophilum*-like 1 was not studied, but we found that the sampled sheep and goats were commonly infested with *R. bursa* (86%), *R. turanicus* (6.6%), *D. marginatus* (6.4%), and very rarely *I. ricinus* (0.06%, only one specimen). Our finding also showed a correlation between *Anaplasma* positivity and the presence of ticks (*p* = 0.0003), compatible with the finding that the prevalence of *A. phagocytophilum*-like 1 was higher in goats infested by ticks than in not infested [7]. Based on the abundance of *Rhipicepahlus* and *Dermacentor* ticks and the very rarity of *I. ricinus*, we can assume that *Rhipicephalus* and *Dermacentor* may play an important role in the transmission of *A. phagocytophilum*-like 1 rather than *I. ricinus*. This assumption is supported by the previous findings that a high prevalence of *A. phagocytophilum*-like variants have been reported in ruminants in the higher semi-arid area of Tunisia, where *I. ricinus* is not present [8]. However, more detailed studies are needed to validate this assumption and to establish what tick species may play a role in the transmission of *A. phagocytophilum*-like 1 in Turkey.

Our sequencing validated RFLP findings, and showed that the sampled small ruminants were found to be infected with *A. phagocytophilum*-like 1. Phylogenetic analysis indicated two main separate branches. The Aplike1OvineCaprine (MT881655) variant obtained in this study, as well as those previously reported from sheep, goats, cattle, and ticks, formed a monophylogenic clade distinct from *A. phagocytophilum* and *A. phagocytophilum*-like 2, and other members of *Anaplasma* spp. [7,8,24].

## 5. Conclusions

This work provides molecular data for the circulation of *A. phagocytophilum-*like 1 for the first time in Turkey. The novel strain is widespread in small ruminants in the Mediterranean area of Turkey with an overall prevalence of 26.5%. This finding revealed that the variant should be considered in the diagnosis of caprine and ovine anaplasmosis.

## Figures and Tables

**Figure 1 animals-11-00814-f001:**
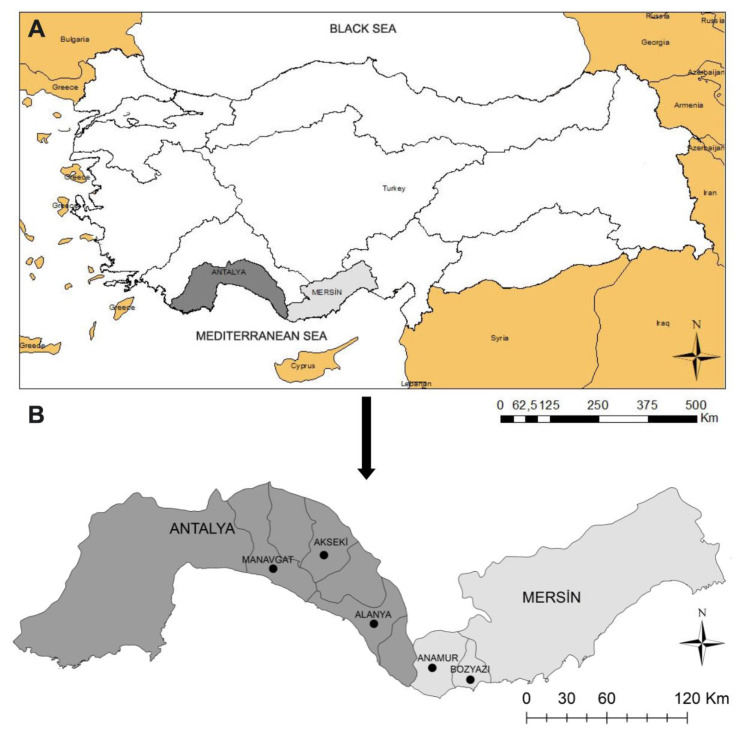
Map of Turkish provinces, indicating the localities studied in the study. (**A**) Geographical position of the provinces of Antalya and Mersin in Turkey. (**B**) Position of localities sampled in the provinces of Antalya and Mersin.

**Figure 2 animals-11-00814-f002:**
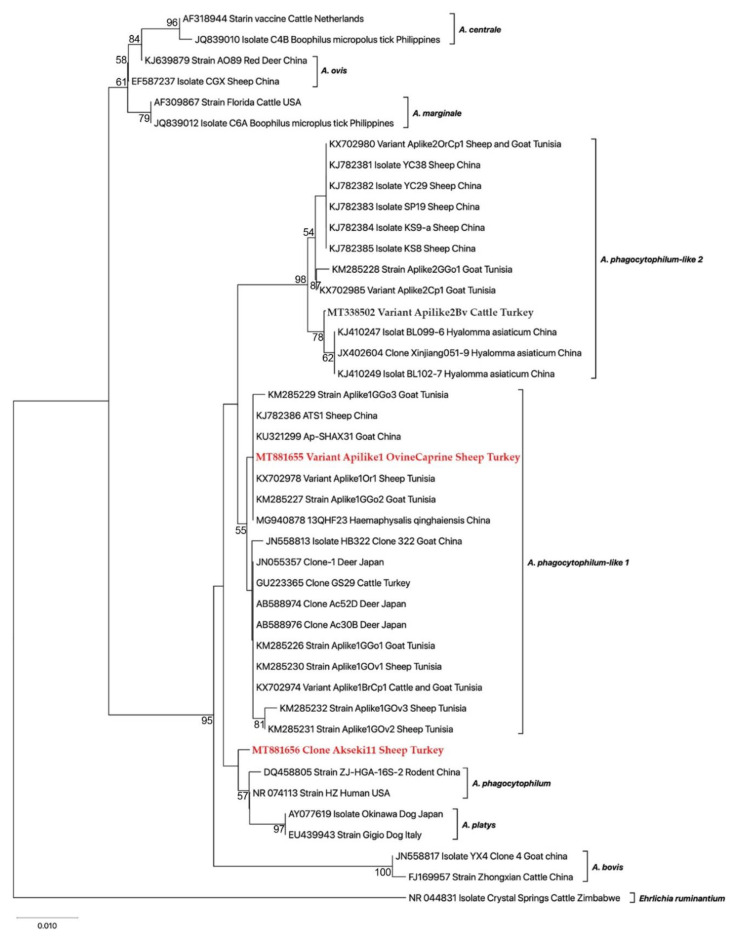
Maximum likelihood phylogenetic tree was inferred with partial sequences (598–599 bp) of the 16S rRNA gene of *Anaplasma* sp. related to *A. phagocytophilum* isolated from sheep in Turkey (highlighted in red) with other *Anaplasma* spp. retrieved from GenBank. Numbers at the nodes refer percentage occurrence in 1000 the bootstrap replication. The new sequences of *A. phagocytophilum*-like 1 and *A. phagocytophilum* from this study were highlighted in red. *Ehrlichia ruminantium* was used as an outgroup.

**Table 1 animals-11-00814-t001:** Oligonucleotide primers used in this study (* Degenerate primer: Y = C or T).

Target Gene	Specificity	Primer Name	Oligonucleotide Dequence (5′-3′)	Annealing	Amplicon Size (bp)	Reference
- - - *16S rRNA*	All *Anaplasma*/*Ehrlichia* - - *A. phagocytophilum* and related variants	EC9 EC12a - SSAP2f SSAP2r	TACCTTGTTACGACTT TGATCCTGGCTCAGAACGAACG - GCTGAATGTGGGGATAATTTAT ATGGCTGCTTCCTTTCGGTTA	54 - 53	1462 - 641–642	[10]
- *groEL*	*-* *A. phagocytophilum*	EphplgroEL(569)F EphgroEL(1142)R - EphplgroEL(569)F EphgroEL(1142)R	ATGGTATGCAGTTTGATCGC TTGAGTACAGCAACACCACCGGAA - ATGGTATGCAGTTTGATCGC TTGAGTACAGCAACACCACCGGAA	54 - - 54	624 - - 573	[23]
*groEL*	*A. phagocytophilum*-like 1	EEGro1F AnaGroe712R AnaGroe240F	GAGTTCGACGGTAAGAAGTTCA ATTAGY *AAGCCTTATGGGTC CCGCGATCAAACTGCATACC	52 - 57	670 - 432	[25]

**Table 2 animals-11-00814-t002:** Samples origin, 16S rRNA PCR, RFLP and groEL PCR.

Host	District/Province	16S rRNA PCR+/No. of Samples	16S rRNA PCR + RFLP			groEL+/16S+	groEL PCR	
-	-	*-*	AP	AP-like 1	AP-like 2		AP	AP-like 1
Goat	Akseki/Antalya	13/56 (23.2%)	2	11	0	12/13	2	10
-	Manavgat/Antalya	26/111 (23.4%)	3	23	0	23/26	3	20
-	Alanya/Antalya	24/55 (43.6%)	0	24	0	24/24		24
-	Anamur/Mersin	15/44 (34.1%)	0	15	0	12/15		12
-	Bozyazı/Mersin	5/30 (16.7%)	0	5	0	5/5		5
Goat Total	-	83/296 (28%)	5	78	0	76/83 (91.5%)	5	71
-	-	-	-	-	-	-	-	-
Sheep	Akseki/Antalya	5/9 (55.6%)	1	4	0	3/5	1	2
-	Manavgat/Antalya	18/103 (17.5%)	0	18	0	17/18	-	17
-	Alanya/Antalya	9/9 (100%)	0	9	0	9/9	-	9
-	Anamur/Mersin	6/16 (37.5%)	0	6	0	5/6	-	5
Sheep Total	-	38/137 (27.7%)	1	37	0	34/38 (89.4%)	1	33
Grand Total	-	121/433 (27.9%)	6 (1.4%)	115 (26.5%)	0	110/115 (95.6%)	6	104

**Table 3 animals-11-00814-t003:** Association of the frequency (16S rRNA PCR) of *Anaplasma phagocytophilum* and related variants in small ruminants with species and tick infestation.

	Species		Presence of Ticks on the Animals	
	Goats *n* (%)	Sheep *n* (%)	No *n* (%)	Yes *n* (%)
Number	296	137	243	190
Positive	83 (28)	38 (27.7)	51 (20.9)	70 (36.8)
Negative	213 (72)	99 (72.3)	192 (79.1)	120 (63.2)
*p*-Value	0.9603		0.0003	

**Table 4 animals-11-00814-t004:** Nucleotide differences among 16S rRNA sequences from *Anaplasma* variants related to *A. phagocytophilum* (598–599 bp).

Host	Genetic Variant ^a^	Country	GenBank	Nucleotide Positions ^b^														Reference
-	-	-	-	823	830	1011	1109	1111	1113	1120	1137	1148	1237	1239	1240	1260	1291	-
Human	Webster	USA	NR_044762	T	T	A	G	T	A	C	A	T	T	T	C	G	C	Unpublished
Horse	Camawi	USA	AF172167	*	*	*	*	*	*	*	*	*	*	*	*	*	*	Unpublished
Dog	Dog2	USA	CP006618	*	*	*	*	*	*	*	*	*	*	*	*	*	*	Unpublished
Deer	Clone 1	Japan	JN055357	C	*	*	*	A	-	*	G	C	*	*	*	*	*	[24]
Goat	Aplike1GGo1	Tunisia	KM285226	C	*	*	*	A	-	*	G	C	*	*	*	*	*	[7]
Goat	Aplike1GGo2	Tunisia	KM285227	C	*	*	*	A	T	*	G	*	*	*	*	*	*	[7]
Sheep	Aplike1GOv1	Tunisia	KM285230	C	*	*	*	A	-	*	G	C	*	*	*	*	*	[7]
Cattle and Goat	Aplike1BvCp1	Tunisia	KX702974	C	*	*	*	A	-	*	G	C	*	*	*	*	*	[8]
Sheep	Aplike1Ov1	Tunisia	KX702978	C	*	*	*	A	T	*	G	*	*	*	*	*	*	[8]
Sheep and Goat	Aplike2OvCp1	Tunisia	KX702980	C	A	G	*	A	T	T	G	C	C	C	T	A	T	[8]
Cattle	Aplike1Bv	Turkey	MT338494	C	*	*	*	A	T	*	G	*	*	*	*	*	*	Unpublished
Sheep and Goat	Aplike1OvineCaprine	Turkey	MT881655	C	*	*	*	A	T	*	G	*	*	*	*	*	*	Present study
Sheep	Aphaakseki11	Turkey	MT881656	*	*	*	*	*	*	*	*	*	*	*	T	*	*	Present study

* Asterisks show the conserved nucleotide positions. ^a^ Applike1OvineCaprine variant has been registered with GenBank under accession number MT881655. ^b^ Numbers indicate the nucleotide position (*A. phagocytophilum*, NC 007797). The position of nucleotide 1011 indicates the substitution of A by G allowing differentiation between variants of *A. phagocytophilum* (like 1 and 2) by *Bsa*I enzyme, while the position of 1137 nucleotide indicates the substitution of A by G allowing for the distinction between *A. phagocytophilum* and related variants (like 1 and 2) by *Xcm*I enzyme [8].

## Data Availability

Data available in a publicly accessible repository.

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
