# Peer review of "Molecular Detection and Phylogeny of Anaplasma phagocytophilum and Related Variants in Small Ruminants from Turkey"

_animals, 2021, doi:10.3390/ani11030814_

Round 1

Reviewer 1 Report

line 34: I would delete "and free-living animals" because TBF is a disease affecting only domestic ones. It sound better: "Anaplasma phagocytophilum is the agent of Tick Borne Fever (TBF) or pasture fever, a disease affecting some species of domestic ruminants (cattle, sheep, goats)

line 72: there's a space between "study" and "secured".

line 122: perhaps the authors wanted to refer  "table 2" not "table 1"

line 188: "..to detect and identify..." not "identity"

Author Response

Reviewer 1

We thank to the reviewer 1 for positive opinion, and her/his valuable comments regarding the our manuscript. All statement (comments, suggestions and corrections) listed by the reviewer 1 were made by tracking change.

line 34: I would delete "and free-living animals" because TBF is a disease affecting only domestic ones. It sound better: "Anaplasma phagocytophilum is the agent of Tick Borne Fever (TBF) or pasture fever, a disease affecting some species of domestic ruminants (cattle, sheep, goats).

Response: The sentence was revised as ‘’Anaplasma phagocytophilum is the agent of Tick Borne Fever (TBF) or pasture fever, a disease affecting some species of domestic ruminants (cattle, sheep, goats)’’ in lines (lines 33-34).

line 72: there's a space between "study" and "secured".

Response: The space between study and scured was deleted (line 73).

line 122: perhaps the authors wanted to refer  "table 2" not "table 1"

Response: Of course, it refers Table 2, it was corrceted, thank for this correction (line 123).

line 188: "..to detect and identify..." not "identity"

Response: It was corrected as ‘’identify’’ (line 186).

From line 54 to 60: pay attention: the font is bolt

Response: It was revised (lines 55-60).

Reviewer 2 Report

Dear authors,
I think that doubts are sufficiently explained to accept the manuscript
Best regards

Author Response

Dear authors,
I think that doubts are sufficiently explained to accept the manuscript
Best regards

Response: We thank to the reviewer 2 for his/her positive approach about our manuscript. 

This manuscript is a resubmission of an earlier submission. The following is a list of the peer review reports and author responses from that submission.

Round 1

Reviewer 1 Report

M and M:

From line 54 to 60: pay attention: the font is bolt

line 59: use "near to the coast" and not close, only to avoid the repetition in the text 

line 64: invert words and numbers "(296 goats, 137 sheep)

In my opinion in this section authors should spend two words regarding blood sampling because they describe well only ticks collection 

line 72/73: I would add "Anaplasma phagocytophilum-like strains DNAs, received from.....were used as positive control"(if I understand well)

line 78: I would add (RFLP) 

line 182: I woud write: "however, it was lower than that observed in other studies conducted in Mediterranean small ruminants" 

line 193-194: this phrase doesn't work so well..I'd delete "with" before the numbers in percentage and I'd put the numbers in brackets...I'd delete "in Turkey" too 

Author Response

Reviewer 1

We thank to the reviewer 1 for positive opinion, and her/his valuable comments regarding the our manuscript. All statement (comments, suggestions and corrections) listed by the reviewer 1 were made by tracking change.

1. From line 54 to 60: pay attention: the font is bolt

Response: It was revised (lines 54-59).

2. line 59: use "near to the coast" and not close, only to avoid the repetition in the text

Response: It was revised (line 58). 

3. line 64: invert words and numbers "(296 goats, 137 sheep)

Response: It was corrected (line 67). 

4. In my opinion in this section authors should spend two words regarding blood sampling because they describe well only ticks collection

Response: We agree with the reviewer that regarding blood sampling, the sentence ‘’Blood samples were drawn from the punctured jugular vein into anticoagulanted (K3-EDTA) vacutainer tubes and stored at -20ºC freezers until DNA extraction’’ was added (lines 67-69).

5. line 72/73: I would add "Anaplasma phagocytophilum-like strains DNAs, received from.....were used as positive control"(if I understand well)

Response: It was revised as ‘’ Anaplasma phagocytophilum-like variants DNAs, received from Dr. Alberto Alberti (University of Sassari, Sassari, Italy) were used as positive controls ’’ (lines 77-79).   

6. line 78: I would add (RFLP)

Response: The RFLB was added (lin 84).

7. line 182: I woud write: "however, it was lower than that observed in other studies conducted in Mediterranean small ruminants" 

Response: It was revised as ‘’…, however, it was lower than that observed in other studies conducted in Mediterranean small ruminants’’… (lines 200-201).

8. line 193-194: this phrase doesn't work so well..I'd delete "with" before the numbers in percentage and I'd put the numbers in brackets...I'd delete "in Turkey" too

Response: Taken into account to the reviewer 2, it was revised as’’… which is not consistent with the previous studies in Turkey that reported values of 66.7% in Central Anatolia [29] and 19.7% in Eastern Anatolia [21]’’ (lines 209-210).

Reviewer 2 Report

This paper describes new data on the presence of A. phagocytophilum-like 1 in sheep and goats in Turkey. The data are sound and clearly presented and the manuscript is of good quality.

However, there are a few issues that should be addressed and they are listed below. The absence of GroEL sequences as well as a more detailed description of ticks from this study are lacking and would greatly improve the manuscript and our knowledge about these variants? They are however not essential and the main result is clearly presented with enough substantial evidence.

Materials and methods

Section 2.1.

Second paragraph : The authors do not describe the blood sampling. A sentence or two should be added L64 to explain blood sampling procedure.

Section 2.2 and 2.4.

A table of primers should be added. Some papers may not be free of charge and easy to get, so informations necessary to conduct experiments should be included in each paper

Section 2.5.

The authors should not use the word a “consensus” sequence for A. phagocytophilum and A. phagocytophilum-like 1, but representative sequences. As we discover later in the results section that all sequences were identical, it's fine.

No GroEL sequences ???

Results

Section 3.1

More details should be given in this section :

  • Tick stages found on sheep should be described : Adults? Sex? Nymphs?
  • Tick species found on sheep and goat separately as they may be different

Section 3.3

Are all A. phagocytophilum also 100% identical??

Table 3 should be improved :

Many nucleotide positions are conserved (830, 1011, 1109 …) so why are they indicated as they give no information??

An A. phagocytophilum-like 2 sequence from database as well as the A. phagocytophilum sequence from this study should be added to highlight differences among A. phagocytophilum, as well as the interest of positions 1011 and 1137. Other conserved positions might become interesting then.

Identical sequences should be gathered and not splitted (KM285227)

Figure 1 : bootstrap values are impossible to read, please enlarge them. More details should be given in the legend about the method, the length of the compared sequences. A figure should have all the necessary informations in the legend for the reader to evaluate the presented results.

Discussion

L198: from the primer sequences and Anaplasma sequences, the authors might know if these A. phagocytophilum “specific” primers could amplify A. phagocytophilum-like variants. So they could give this information.

L200-203: this part related to the vector should not be placed here in the middle of the discussion about primers specifity, but added in the following part starting L208.

L218: Tick presence has been observed only once (at the time of blood sampling supposedly) in this study, while Anaplasma is a long term infection. So only one tick collection is not sufficient to certify the absence of ticks on animals all year around. And tick species may differ according to the seasons. Do the authors have informations about the local presence/absence/species of ticks in the different prospected regions??  

Minor comments

L33 : (english error) tick-borne fever does not belong to the genus Anaplasma, but the causal agent. Please rephrase

L34 : (english error and scientific error) The bacterium is a pathogenic species, not the highly

L40, L44, L48, L50 and throughout the whole document: the use of the word variants instead of strains seems more appropriate

L40 : documented not decumented

L56/57 : representative of the Mediterranean region

L71 : please rephrase to “genomic DNA from blood of clinically….”

L164 : please rephrase

L170 : a survey was carried out to detect and identify A. phagocytophilum…

L192-194 : please rephrase : … consistent with previous studies in Turkey that reported values of 66.7% in Central Anatolia and 19.7% in Eastern Anatolia.

L223: please change “is absent”.

Author Response

Reviewer 2

This paper describes new data on the presence of A. phagocytophilum-like 1 in sheep and goats in Turkey. The data are sound and clearly presented and the manuscript is of good quality.

However, there are a few issues that should be addressed and they are listed below. The absence of GroEL sequences as well as a more detailed description of ticks from this study are lacking and would greatly improve the manuscript and our knowledge about these variants? They are however not essential and the main result is clearly presented with enough substantial evidence.

Response: We thank to the reviewer 2 for his/her positive approach about our manuscript. We agree with the reviewer that absence of GroEL sequences could be the weakness of this article. To date no information is available on the circulation on A. phagocytophilum-like variants in Turkey. Therefore, this study aimed to investigated the presence and prevalence of A. phagocytophilum and related variants in small ruminants by 16S rRNA and restriction enzyme digestion in sampling sites.

Regarding the description of collected ticks, the phrases ‘’A total of 1475 adult ticks (449 females, 1026 males) were collected from goats (1409/1475, 95.5%) and sheep (66/1475, 4.5%). Six tick species were identified among all collected ticks. Rhipicephalus bursa (1269/1475, 86%) was the dominant tick species, followed by R. turanicus (98/1475, 6.6%), Dermacentor marginatus (94/1475, 6.4%), Hyalomma marginatum (8/1475, 0.5%), R. sanguineus s.l. (5/1475, 0.3%), and Ixodes ricinus (0.06%, only one specimen). The goats were infested with all the identified tick species, whereas sheep were infested with R. bursa and R. turanicus was added in lines 113-119.

Materials and methods

Section 2.1.

Second paragraph : The authors do not describe the blood sampling. A sentence or two should be added L64 to explain blood sampling procedure.

Response: We thank to the reviewer 2 for this suggestion, the sentence ‘’Blood samples were drawn from the punctured jugular vein into anticoagulanted (K3-EDTA) vacutainer tubes and stored at -20 oC freezer until DNA extraction’’ was added (lines 67-69).

Section 2.2 and 2.4.

A table of primers should be added. Some papers may not be free of charge and easy to get, so informations necessary to conduct experiments should be included in each paper

Response: We thank to the reviewer 2 for this suggestion, a descriptive primer lists used in this study was presented as a new Table (Table 1) (lines 130-131).

Section 2.5.

The authors should not use the word a “consensus” sequence for A. phagocytophilum and A. phagocytophilum-like 1, but representative sequences. As we discover later in the results section that all sequences were identical, it's fine.

Response: The word “consensus” was changed as ‘’representative’’ (line 100).

No GroEL sequences ???

Response: As mentioned above, we agree with the reviewer 2 that absence of GroEL sequences could be the weakness of this article. However, to date no information is available on the circulation on A. phagocytophilum-like variants in Turkey. Therefore, this study aimed to investigated the presence and prevalence of A. phagocytophilum and related variants in small ruminants by 16S rRNA and restriction enzyme digestion in sampling sites.

Results

Section 3.1

More details should be given in this section:

Tick stages found on sheep should be described : Adults? Sex? Nymphs?

Tick species found on sheep and goat separately as they may be different

Response: As mentioned above, more details regarding the description of collected ticks was added as ‘’A total of 1475 adult ticks (449 females, 1026 males) were collected from goats (1409/1475, 95.5%) and sheep (66/1475, 4.5%). Six tick species were identified among all collected ticks. Rhipicephalus bursa (1269/1475, 86%) was the dominant tick species, followed by R. turanicus (98/1475, 6.6%), Dermacentor marginatus (94/1475, 6.4%), Hyalomma marginatum (8/1475, 0.5%), R. sanguineus s.l. (5/1475, 0.3%), and Ixodes ricinus (0.06%, only one specimen). The goats were infested with all the identified tick species, whereas sheep were infested with R. bursa and R. turanicus’’ (lines 113-119).

Section 3.3

Are all A. phagocytophilum also 100% identical??

Response: Yes, all A. phagocytophilum also 100% identical.

Table 3 should be improved:

Many nucleotide positions are conserved (830, 1011, 1109 …) so why are they indicated as they give no information??

An A. phagocytophilum-like 2 sequence from database as well as the A. phagocytophilum sequence from this study should be added to highlight differences among A. phagocytophilum, as well as the interest of positions 1011 and 1137. Other conserved positions might become interesting then.

Response: We agree with the reviewer, thank for this comment. An Anaplasma phagocytophilum-like 2 sequence from database (Aplike2OvCp1 identified from sheep and goat from Tunisia, GenBank accessiono. KX702980) and the A. phagocytophilum sequence obtained in this study (Aphaakseki11 identified from sheep, GenBank accessio no. MT881656) were added to the revised Table (Table 4) (lines 161-166).

Figure 1 :bootstrap values are impossible to read, please enlarge them. More details should be given in the legend about the method, the length of the compared sequences. A figure should have all the necessary informations in the legend for the reader to evaluate the presented results.

Response: We agree with the reviewer that bootstrap values are impossible to read, please enlarge them. We thank for this suggestion. In the revised figure, the bootstrap values were enlarged. Also more details were given in the revised legend of Fig. 2 as ‘’ Phylogenetic tree inferred with partial sequences (598–599 bp) of the 16S rRNA gene of Anaplasma sp. closely related to A. phagocytophilum isolated from sheep in Turkey (highlighted in red) with other Anaplasma spp. retrieved from GenBank using the maximum likelihood method. Numbers at the nodes represent the bootstrap values with 1000 replicates (only percentages greater than 50% were presented). The sequences were given as GenBank accession number, the strain or isolate name, host or vector and country. The novel sequences of A. phagocytophilum-like 1 and A. phagocytophilum in red type were obtained in this study. Scale bar represents 0.01 substitutions per nucleotide position. Ehrlichia ruminantium (GenBank number: NR_044831) was used as an outgroup in the tree’’ (lines 169-178).

Discussion

1. L198: from the primer sequences and Anaplasma sequences, the authors might know if these A. phagocytophilum “specific” primers could amplify A. phagocytophilum-like variants. So they could give this information.

Response: We agree with the reviewer, thank for this correction, the statement was corrected as ‘’ However, the EE1/EE2 primers are universal for the detection of all Anaplasma spp. including A. phagocytophilum-like variants [32]. It has been also reported that the SSAP2f/SSAP2r can amplify not only A. phagocytophilum, but also A. phagocytophilum-like variants [7,8,24]’’ (lines 213-216).

2. L200-203: this part related to the vector should not be placed here in the middle of the discussion about primers specifity, but added in the following part starting L208.

Response: We thank to the reviewer 2 for this suggestion, this part was removed and added to the following part starting line 221.

3. L218: Tick presence has been observed only once (at the time of blood sampling supposedly) in this study, while Anaplasma is a long term infection. So only one tick collection is not sufficient to certify the absence of ticks on animals all year around. And tick species may differ according to the seasons. Do the authors have informations about the local presence/absence/species of ticks in the different prospected regions??

Response: We agree with the reviewer that so only one tick collection is not sufficient to certify the absence of ticks on animals all year around, and tick species may differ according to the seasons. However, to date no data about Anapalsma phagocytophilum-like varants in Turkey. Furthermore, no data on the ixodid tick species and distribution in small ruminats were reported in the sampled region (Mersin and Antalya). Therefore, we have not informations about the local presence/absence/species of ticks in the different prospected regions, where Anaplasma phagocytophilum-like varants occur.

Minor comments

1. L33 : (english error) tick-borne fever does not belong to the genus Anaplasma, but the causal agent. Please rephrase

Response: It was changed as ‘’ Anaplasma phagocytophilum is the agent of Tick borne fever (TBF) or pasture fever, a disease affecting domestic and free-living ruminants’’ (lines 33-34).

2. L34 : (english error and scientific error) The bacterium is a pathogenic species, not the highly

Response: It was revised as’’The bacterium is a pathogenic species…’’ (line 34).

3. L40, L44, L48, L50 and throughout the whole document: the use of the word variants instead of strains seems more appropriate

Response: The word variant (s) instead of strain (s) were used in throughout the whole document.

4. L40 : documented not decumented

Response: It was changed as documented (line 40).

5. L56/57 : representative of theMediterranean region

Response: It was deleted, and the sentences were revisea as ‘’ This survey was conducted in small ruminants farmed in three districts (Alanya, Akseki, Manavgat) from Antalya (latitude 36° 53′ N, longitude 30° 42′ E) and two districts (Anamur, Bozyazı) from Mersin (latitude 36º 47′ N, longitude 34º 37′ E) provinces of Turkey (Fig. 1)’’ (lines 54-56).

6. L71 : please rephrase to “genomic DNA from blood of clinically….”

Response: It was changed as genomic DNA from blood of clinically… (line 77).

7. L164 : please rephrase

Response: The sentence was changed as ‘’ The genetic diversity of A. phagocytophilum is much greater than expected’’ (line 182).

8. L170 : a survey was carried out to detect and identify A. phagocytophilum…

Response: It was revised as suggested (lines 187-188).

9. L192-194 : please rephrase : … consistent with previous studies in Turkey that reported values of 66.7% in Central Anatolia and 19.7% in Eastern Anatolia.

Response: It was revised as ‘’ …which is not consistent with the previous studies in Turkey that reported values of 66.7% in Central Anatolia [29] and 19.7% in Eastern Anatolia’’ (lines 209-210).

10. L223: please change “is absent”.

Response: It was changed as’’ …where I. ricinus is not present (line 239).

Reviewer 3 Report

Title: "Molecular detection and phylogeny of Anaplasma pahagocytophilum and related strains in small ruminants from Turkey.”

This manuscript is well written, easy to understand and introduce new knowledge over Anaplasma in small ruminants.

Points to review:

Material and Methods:

Line 54-57: “... small ruminants farmed in three districts (Alanya, Akseki, Manavgat) from Antalya ... and two districts (Anamur, Bozyazi) from Mersin ... provinces, representative of Mediterranean region of Turkey.” I think that a map to understand what representative of Mediterranean region of Turkey means should help to readers.

Lines 61-63:

“The sample size was calculated using the online tool Sample Size Calculator (www.calculator.net/sample-sizecalculator. html), for a confidence level (CL) of 95%, an error margin of 5%“. The URL really is www.calculator.net/sample-size-calculator.html, but more important, this equation must include expected prevalence. In case we have no previous idea, this number in general for infinite population is 384; in case that authors take into account their last manuscript in cattle, prevalence could be an approximation to 32% (Aktas et al, 2021), and in this case the number is 335, but 384 or 335, it should be 384/335 goats and 384/335 sheep, because hosts are different and in different conditions… Or not? Authors must explain why 296 goats and 137 sheep.

Lines 73:

Dr Alberto, I suppose there must be Dr. Alberti (from complete name Alberto Alberti to Dr. Alberti, and also in Acknoledgements and Reference 17

Results:

Line 132:

Authors said: “To validate the RFLP results and identify genetic variants of A.phagocytophilum-like 1, randomly selected 10 representative samples were sequenced.” Why 10? If they are not using a specific PDR to A.phagocytophilum-like 1, why are they confirming only 10 from 104? Authors must explain why 10 and not 104.

Discussion:

Lines 183-186:

“These findings show that biotic (abundance of reservoirs and intensity of tick vectors) and abiotic (geographic location, husbandry practices, air temperature, relative humidity, vegetation type) factors may affect the distribution and frequency of anaplasmosis. “ these findings are findings of other people. This manuscript does not include any data to confirm or not this access, so I think this phrase must be eliminated or better explain in base to other works, as it cannot be obtained in this study.

Author Response

Reviewer 3

Title: "Molecular detection and phylogeny of Anaplasma pahagocytophilum and related strains in small ruminants from Turkey.”

This manuscript is well written, easy to understand and introduce new knowledge over Anaplasma in small ruminants.

Response: We thank to the reviewer 3 for positive opinion, and her/his valuable comments regarding the our manuscript.

Points to review:

Material and Methods:

Line 54-57: “... small ruminants farmed in three districts (Alanya, Akseki, Manavgat) from Antalya ... and two districts (Anamur, Bozyazi) from Mersin ... provinces, representative of Mediterranean region of Turkey.” I think that a map to understand what representative of Mediterranean region of Turkey means should help to readers.

Response: A map of Turkish provinces as Fig. 1, indicating the localities studied in the study. (A) Geographical position of the provinces of Antalya and Mersin in Turkey. (B) Position of localities sampled in the provinces of Antalya and Mersin was added (lines 73-80).

Lines 61-63:

“The sample size was calculated using the online tool Sample Size Calculator (www.calculator.net/sample-sizecalculator. html), for a confidence level (CL) of 95%, an error margin of 5%“. The URL really is www.calculator.net/sample-size-calculator.html, but more important, this equation must include expected prevalence. In case we have no previous idea, this number in general for infinite population is 384; in case that authors take into account their last manuscript in cattle, prevalence could be an approximation to 32% (Aktas et al, 2021), and in this case the number is 335, but 384 or 335, it should be 384/335 goats and 384/335 sheep, because hosts are different and in different conditions… Or not? Authors must explain why 296 goats and 137 sheep.

Response: We thank to the reviewer 3 for the comments, but we do not agree with the reviewer. In this study, it was aimed to determine the prevalence of small ruminant anaplasmosis in the sampled region, not separately ovine anaplasmosis or caprine anaplasmosis. Yes, hosts are different, but their conditions are not different because goats and sheep sampled in this study graze together in the same herd, and the same conditions. Regarding why 296 goats and 137 sheep, in this region, sheep constitute about %30 of the small ruminant population. Therefore, the number of goat samples was taken more. The sample size was calculated based on statistical and epidemiological concepts, considering the population of small ruminants in the area as accepted error of 5%, at 95% level of confidence. Due to the lack of any information about the prevalence of small ruminant anaplasmosis in the sampled region, it was assumed an infection prevalence of 50%.

Lines 73:

Dr Alberto, I suppose there must be Dr. Alberti (from complete name Alberto Alberti to Dr. Alberti, and also in Acknoledgements and Reference 17

Response: Complete name Dr. Alberto Alberti was added as ‘’Anaplasma phagocytophilum-like variants DNAs, received from Dr. Alberto Alberti (University of Sassari, Sassari, Italy) were used as positive controls’’ (lines 97-104).

Results:

Line 132:

Authors said: “To validate the RFLP results and identify genetic variants of A.phagocytophilum-like 1, randomly selected 10 representative samples were sequenced.” Why 10? If they are not using a specific PDR to A.phagocytophilum-like 1, why are they confirming only 10 from 104? Authors must explain why 10 and not 104.

Response: The nested primers (SSAP2f/SSAP2r) used in this study do not species-specific, it can amplify not only A. phagocytophilum, but also A. phagocytophilum-like variants (A. phagocytophilum-like 1 and A. phagocytophilum-like 2). To discriminate among A. phagocytophilum, A.phagocytophilum-like 1, and A. phagocytophilum-like 2, species-specific diagnostic approach based on16S rRNA PCR combined to restriction enzymes (RFLP) were performed in the study. Then, ten randomly selected isolates (positive for A.phagocytophilum-like 1) were sequenced to validate the RFLP results, and more importantly to perform an A. pahocytophilum like 1 genetic comparison. We agree with the reviewer that ten sequences could be the weakness of this article,  but sequencing 104 isolates would have been very costly. There is not enough allocation in the project budget for this.

Discussion:

Lines 183-186:

“These findings show that biotic (abundance of reservoirs and intensity of tick vectors) and abiotic (geographic location, husbandry practices, air temperature, relative humidity, vegetation type) factors may affect the distribution and frequency of anaplasmosis. “ these findings are findings of other people. This manuscript does not include any data to confirm or not this access, so I think this phrase must be eliminated or better explain in base to other works, as it cannot be obtained in this study.

Response: We thank to the reviewer 3 for this suggestion, we agree with the reviewer that“ this manuscript does not include any data to confirm or not this access’’. Therefore, this phrase was deleted from the text (line 280).
